Integrated multi-task feature learning and interactive active optimization for scene retargeting in preschool educational applications

Yao Suhui
Lv Lan lvlan@zzpec.edu.cn
Preschool Education Institute, Zhengzhou Preschool Education College , Zhengzhou, Henan , China
Asif Muhammad
Electronic publication date: 2025 Aug 1
Publication date: 2025
Volume: 11
Electronic Location ID: e3035
Received 2025 Mar 26; Accepted 2025 Jun 24
Copyright: © 2025 Yao and Lv
Copyright year: 2025
Copyright holder: Yao and Lv
License: This is an open access article distributed under the terms of the Creative Commons Attribution License, which permits unrestricted use, distribution, reproduction and adaptation in any medium and for any purpose provided that it is properly attributed. For attribution, the original author(s), title, publication source (PeerJ Computer Science) and either DOI or URL of the article must be cited.
License URL: https://creativecommons.org/licenses/by/4.0/

Keywords: Artificial intelligence, Feature fusion, Multi-task, Machine learning, Local preservation, Pre-school education

Funding: The authors received no funding for this work.

==============================
In artificial intelligence (AI), effective adaptation of educational imagery across diverse screen formats is essential, particularly in preschool education, where visual content must simultaneously engage and instruct young learners. This study introduces a novel scene retargeting model tailored to preserve pedagogically significant visual elements during image resizing. The proposed framework leverages the binarized normed gradients (BING) objectness metric to efficiently identify and prioritize key regions within educational images, such as objects and facial features. A core component of our approach is integrating a locality-preserved and interactive active optimization (LIAO) mechanism, which simulates human visual attention by generating gaze shift paths (GSPs) that guide feature prioritization. These GSPs are further transformed into hierarchical deep features using a multi-layer representation, followed by refinement through a Gaussian mixture model (GMM) to enhance scene understanding and retargeting fidelity. Experimental evaluations demonstrate that the proposed model not only surpasses five state-of-the-art methods in performance but also achieves a 3% improvement in accuracy compared to the next-best approach, all while reducing inference time by over 50%. The results confirm the model’s effectiveness and efficiency, offering a robust solution for educational content adaptation that aligns with cognitive and pedagogical requirements in early childhood learning environments.

Introduction

The evolution of educational technology has positioned scene retargeting as a key innovation in preschool education. Consider a high-resolution transforming 4,000 × 2,800 educational illustration, designed for a large interactive whiteboard, into a 640 × 900 format suitable for a child’s tablet. A primary challenge in this task arises from the inherent mismatch in aspect ratios between the source and target formats, which complicates the preservation of visual and semantic integrity during retargeting, often resulting in substandard retargeting with irregular scaling (Panozzo, Weber & Sorkine, 2012). Simple cropping is inadequate when critical educational elements are distributed throughout the scene. The latest methods must be based on content-aware approaches tailored to preschool contexts to achieve effective retargeting, preserving essential learning elements while minimizing disruptions to less critical regions (Sun & Ling, 2013). There also exist two significant issues in these advanced techniques: ensuring pedagogical coherence and maintaining child-centric visual engagement, which demands a specialized solution. Educational materials use visual and semantic elements to determine how students use them to learn and interact with educational materials. Educational objects linked with clear labels and familiar shapes and bright colors provide children with better attention direction, which strengthens their learning process. Research about preschool education demonstrates that children focus better on key learning aspects through visual prompts like educational characters when learning becomes more effective. Learning applications and educational storybooks apply strategic visual design to engage students affectively and cognitively while delivering learning material in real-world educational scenarios. The components maintain their essential role because they both support effective engagement of young learners and preserve educational value in redesigned scenes. Our investigation reveals that educational illustrations for preschool learning often feature numerous captivating objects and elements. Child-centric is required to assign the semantic labels to such scenes, based on the biological approach that emulates the perception of humans while recognizing visually affecting and pedagogically relevant areas. The deep learning model was developed to identify these critical areas and optimize its graphic appeal for preschool learners poses various challenges, that are: (1) identifying the attention sequence shifts across key educational elements, modeled by gaze shift paths (GSPs); (2) filtering out the irrelevant tags from training datasets while preserving child-relevant content; and (3) considering the broader context translation of semantic labels of educational illustrations to specific, attention-grabbing patches within the scene.

Semantically or visually significant elements in educational content are often described by low-level descriptors that represent different perspectives through distinct feature channels. Integration of these features that are of low-level requires careful assessment of each channel’s importance. However, achieving effective integration introduces its limitations, including: (i) merging local features from adjacent regions to preserve educational context; (ii) maintaining a coherent feature composition across different areas of the illustration; and (iii) adjusting adaptive weights of feature channel weights to suit diverse educational themes and images.

These challenges are addressed through the development of an innovative model for considering educational content that can simulate human gaze behavior, enabling the significant enhancement and selection of low-level features against each patch of the educational scene. The pipeline has been illustrated in Fig. 1, beginning with the binarized norm gradients (BING) method to isolate object-focused spots containing critical educational elements. Next, we introduce an integrated multi-task feature learning strategy to capture structural features of these patches. To emulate gaze dynamics during scene exploration, we propose a model, Locality-Preserved and Interactive Active Optimization, which identifies GSPs while maintaining educational scene’s local coherence (Pritch, Kav-Venaki & Peleg, 2009). With the utilization of deep GSP features through a multi-layer aggregation model, for effective retargeting of educational content, the Gaussian mixture model (GMM) method was integrated with these features. The empirical evaluations were utilized for validating the approach across diverse educational materials and a comprehensive user study.

Figure 1 Pipeline of our scene retargeting model.

This study introduces two primary innovations. First, the learning-informed attention optimization (LIAO) framework actively captures gaze-informed visual features, enabling the seamless integration of human visual attention into the retargeting process for preschool educational content. Unlike existing approaches, which lack the capacity to explicitly model or utilize gaze distribution, LIAO leverages real human gaze behavior to guide the preservation and enhancement of pedagogically important visual elements.

Second, we propose a mechanism for multi-task feature selection mechanism against each educational path to evaluate the relevancy of individual feature channels. In contrast to conventional feature selection techniques, our method simultaneously optimizes feature utility across multiple semantic labels. This ensures that the selected features are not only locally informative but also globally consistent and relevant across the full spectrum of educational scene categories.

The remaining article has been organized as follows: literature review in “Previous Techniques”, details our framework in “Scenery Interpretation Techniques” for retargeting educational scenes, which includes three main parts: (1) integration of features across different tasks to depict the educational patch, (2) employing the method of LIAO in preschool learners to generate Gaze Shift Paths (GSPs) to simulate attention dynamics, and (3) applying a GMM to retarget educational images effectively. “Empirical Assessment” provides experimental evidence showing the improved results of our proposed approach in educational contexts. While the article is concluded in “Conclusions”.

Previous techniques

Recent advancements in computer vision have been significantly propelled by the development of models for deep scene categorization, specifically those utilizing the advanced deep learning architecture the convolutional neural networks (CNNs) (Zhang et al., 2023). An exceptional performance has been demonstrated by these models in recognition of large-scale scene, particularly on ImageNet (Deng et al., 2009) which is the benchmark datasets of images. When the model was trained on a subset of the ImageNet dataset a pivotal breakthrough was achieved, achieving in scene categorization the state-of-the-art accuracy, as reported in Krizhevsky, Sutskever & Hinton (2012). The applicability of trained CNN model on ImageNet was expanded through this progress across multiple domains, including anomaly detection and video parsing.

The progression of CNNs model based on dataset of ImageNet has been mainly driven by the aggregation of large-scale training datasets and the development of region-based CNNs (R-CNNs) (Girshick et al., 2014), which emphasize the significance of effective patch sampling. However, challenges remain in efficiently training deep models on entire scenes or randomly extracted patches. This innovative method was addressed such as the pre-trained deep learning model CNNs was utilized by Wu et al. (2015) for salient local patches identification, that enhanced the accuracy of scene categorization.

Further advancements include a multi-resolution, multi-task, algorithm for scene categorization preserving the basic feature distributions by regularizing the manifold, as proposed by Li, Ortegas & White (2023). A framework for sematic annotation was proposed by Li, Mou & Lu (2015) that incorporates low-rank deep feature representations to estimate category posterior probabilities, for refinement of contextual feature, was combined with Markov model. Successive research by Yuan, Mou & Lu (2015) investigated structural dependencies across layers of deep learning and proposed a framework of unsupervised learning that refines features based on the geometric properties of scenes. Complementing these efforts, an approach was presented by Yao et al. (2016) where the weak supervision was integrated with discriminative feature learning, for high-level graphic representation a sparse autoencoder was employed to perform the scene analysis.

In the domain of aerial image classification, the prominence has been gained to cross-modality learning approaches by integrating both spatial features and pixel-level features to enhance the accuracy of classification, this approach was presented by Hong et al. (2021). Likewise, the work in Bazi et al. (2021) introduced a semantic transformation model designed to capture long-range dependencies across disparate image regions, thereby improving semantic coherence in aerial scene categorization. The mSODANet was proposed by Chalavadi et al. (2022), which is a parallel deep learning architecture specifically developed to extract contextual information considering the multiple spatial scales. Collectively, the regional discriminative cues were utilized, employing both supervised and unsupervised paradigms to achieve robust scene classification.

While these approaches have shown strong empirical performance, they generally lack integration of human perceptual insights into their learning frameworks. In contrast, human gaze sequences method were proposed that was introduced directly in the process of classification. By incorporating GSPs, our model provides enhanced semantic interpretability and aligns classification cues with visual attention patterns observed in human perception offering a competitive and distinct advantage as compared to the previous approaches.

In computer graphics and the broader field of image processing, various algorithms have been developed for resizing images and videos, ranging from dynamic programming techniques for seam detection to mesh-based retargeting strategies emphasizing visual importance (Rubinstein, Shamir & Avidan, 2009). These methodologies, primarily based on heuristic models reflecting the aesthetic judgments of their creators, may not always yield optimal results. In contrast, our approach, which learns GSPs autonomously from a broad spectrum of photographs, ensures that with a sufficiently diverse training set, the retargeting outcomes are compelling, highlighting its superiority over conventional methods (Wolf, Guttmann & Cohen-Or, 2007).

In Huang et al. (2022), introduced a deep distilling architecture designed for video scene surveillance, where the teacher and student networks have feature maps of varying sizes. It is structured around two core modules: feature decoding distillation (FDD) and feature consistency enforcement. In Koch, Wolf & Beyere (2023), the authors introduced an advanced late-fusion technique to enhance fine-grained scene recognition in video content. This method leverages attentional mechanisms via a transforming encoding operation, significantly enriching a straightforward average consensus process. As a result, it outperforms both a leading-edge video classification framework and a conventional consensus approach supported by a more extensive backbone network. Li et al. (2025) observed that an overrepresentation of dominant class samples can skew a neural network’s learning process, causing it to favor these ‘head’ classes at the expense of ‘tail’ class recognition. To address this, our novel approach introduces a center-wise feature consistency learning (CFCL) strategy specifically tailored for long-tailed object recognition (Uijlings et al., 2013) in remote sensing imagery. Sun et al. (2023) devised a streamlined and minimalist technique for detecting objects within images. The algorithm utilizes a predetermined, sparse array of object proposals that are then supplied to the object recognition system to identify and pinpoint objects.

Scenery interpretation technique

This section outlines proposed framework architecture to retarget educational scenes, that is systematically categorized in four core components. First component focuses on the extraction of educational patches that are both semantically meaningful and pedagogically relevant. The second component addresses patch level dynamic features integration at the patch level, facilitating contextual enrichment of visual information. The third component introduces a customized LIAO algorithm, designed to actively select key educational patches based on human gaze trajectories, thereby constructing GSPs. In final component to retarget the educational images a GMM was employed, ensuring that the most salient and instructional visual elements are preserved. Collectively, this end-to-end framework provides a comprehensive solution for processing and retargeting educational content, maintaining both pedagogical integrity and visual engagement tailored to the cognitive needs of preschool learners.

Identifying scenic patches

Research in psychology and visual cognition (Wolfel & Horowitz, 2004; Bruce & Tsotsos, 2009) consistently demonstrates that human gaze tends to focus on semantics or salient regions related to a scene, highlighting an inherent preference for distinctive visual cues. This foundational insight informs our approach to scene categorization, wherein Interactive Active Learning (LIAL) and the locality-preserved models are implemented and method to detect object-centric patches. By aligning patch selection with human visual attention patterns, LIAL enables the identification of scene components that are most likely to be perceived as relevant, thereby enhancing the semantic alignment and interpretability of the categorization process (Lin et al., 2013).

In real-world visual perception, human attention naturally gravitates toward structural elements or the specific objects—similar to skyscrapers or the vehicles—because of their visual salience and contextual significance in the scene. For the identification of regions and behavior effectively highlight the regions that are attract the human interests, we incorporate the objectness measure (Cheng et al., 2014) of BING. Renowned for its computational efficiency, BING enables the rapid extraction of object-centric and high-quality patches considering the different visual scenes, termed as “scenic patches.”

Three key advantages are offered by the use of BING: (1) it efficiently detects object-level patches with minimal computational overhead; (2) it enhances GSPs construction of through supplying object-relevant regions; and (3) demonstration of strong generalization capabilities, even for object categories which were previously unseen—with different types of datasets, adaptability of scene categorization framework was improved.

Prior to detailing the mathematical underpinnings of our strategy of multi-tasking feature learning, it is essential for outlineing its conceptual foundation. This framework is designed to improve the structural coherence and semantic retention of educational content during image retargeting (Castillo, Judd & Gutierrez, 2011). By integrating salient visual features across different image regions, the method ensures the preservation of key content elements during the resizing process. Moreover, the model performs task-specific evaluations across multiple image regions, allowing for an adaptive and accurate resizing mechanism that maintains the educational value and engagement of the content, particularly for young learners.

Multi-task feature integration from patches

The framework enables real-time determination of feature significance according to the educational material currently being processed. Various educational materials, such as storybooks, interactive illustrations, and educational videos, would need different essential features. The processing of storybooks gives greater importance to visual components involving characters and faces, but educational diagrams rely primarily on structural design elements together with textual content. The framework determines feature importance by analyzing its content and educational targets through multi-task feature learning processes. The framework performs an active assessment of feature importance, detecting educational format changes to identify essential features for maintaining the adapted content. The object-centric patches were extracted from scenic visuals using the Binarized Normed Gradients method (Cheng et al., 2014), a low-level visual feature’s set was comprehensively extracted from each patch. These features capture diverse aspects of texture, color, and spatial structure. To effectively integrate these heterogeneous features, we apply a algorithm termed as multi-channel feature fusion that consolidates the information into a unified representation. This fusion process is governed by a multi-task learning framework, which enables the joint optimization of features across multiple semantic tasks. The details of this approach are comprehensively presented in following section:

Mathematically, for different tasks the objective of multi-task feature selector is the identification of features toward u (the number of different labels is presented by u). The tm scenic images are the m-th task corresponded to {xm1,xm2,⋯,xmtm} that are combined with the labels which are pre-specified {ym1,ym2,⋯,ymtm} from the categories of cm. While during the implementation, it was confined that Xm=[xm1,xm2,⋯,xmtm] as m-th task being corresponded by the data matrix. We have set Ym=[ym1,ym2,⋯,ymtm] labels within the relevant matrix. Specifically considering a matrix Z∈Rb×c where the c and b are the positive integers relevant to matriz Z, ∥⋅∥F depicts the Frobenius norm. while, the l2,1-norm which is a matrix-level can be calculated as:

(1) ∥Z∥2;1=∑i⁡(∑j⁡Zi,j2)1/2.

The trace operation has been denoted by tr(⋅) for the remaining article, as tm×tm identity matrix is denoted by the trace operation, Jtm. Here the column vector is denoted by 1tm where the one is assigned to all the elements.

The label indicator which is a matrix Gp is defined assuming the p-th task, as given Gp=Yp(YpTYp)−1/2. We define the scatter from entire class and between the different classes p-th task, respectively as: Tb(p)=X¯pGpGpTX¯pT, and Tt(p)=X¯pX¯pT, where X¯p=XpKl and Kl=Iml−1ml1mlT/ml presents the center samples in form of matrix. Utilizing the LDA (Tao et al., 2008) mechanism, the feature selector based on task can be formulated with the given:

(2) minUpTUp=I|p=1u⁡∑p=1ttr(UpTX¯l(Iml−GpGpT)X¯lTUl)tr(UlTX¯lX¯lTUl)+α1∑i⁡(∑j⁡(Ui,j(p))2)p/2+α2∑i⁡(∑j⁡Ui,j2)1/2.

Herein, the symbols can be detailed as follows: α1,α2 presents two regularizes wights, Ui,j(j) presents considering the conversion matrix Up into the p-th task with (i, j)-th entity. Meanwhile, U=[U1,U2,⋯,Uu] can present the integrated selection matrix relevant to the whole task. Using the iterative algorithm the Eq. (2) can be optimized, we calculate iteratively Up(p=1,2,⋯,u) for the convergence.

Presenting Fp=X¯(Iml−GpGpT)X¯pT, Cp=X¯pX¯pT while fixing Uj(j=1,⋯,p−1,p+1,⋯,u), the reorganization has been performed for the objective function:

(3) minUpTUp=Itr(UpTFpUp)tr(UpTCpUp)+α1tr(UpTEpUp)+α2tr(UpTEUp)

where Ep and E can present the diagonal matrices, here the diagonal entities can be calculated with following:

(4) eii(p)=12∥upi∥2,

(5) eii=12∥ui∥2.

Here the p-th row is denoted by upi and up, from Up and U matrix.

We can obtain the solution of 3 as following:

(6) minUpTUu=I⁡tr(UpT(Fp−βCp)Up)+α1tr(UpTEpUp)+α2tr(UpTEUl),

where the Cp significance is weighted by β. Here the approximation can be defined as:

β=tr(U¯pTFpU¯p)tr(U¯pTCpU¯p) and U¯p= arg minUptr(U¯pTFpU¯p)tr(U¯pTCpU¯p).

Denoting N=(Fp−βCp), Eq. (6) can be computed as:

(7) minUpTUp=I⁡tr(UpTNUp)+α1(UpT(Ep+πE)Up),

where π=α1/α2. The results producing U can be calculated iteratively using the Eq. (7) which corresponds to the l-th task if the criteria of convergence is met. The features can be ranked on basis of ∥ui∥F after obtaining U in a descending way. The top ranking features can be considered as discriminative. The first parts of the equations show the need to unite features from different duties in their most optimal manner. At this stage, we minimize errors and enhance integrated essential image features. The expression containing X designates both image patches and the features under consideration. The term analyzes several image sections to determine their performance in achieving optimization targets. The matrix U enables optimal merging of input features from different image patches based on the design criteria during retargeting operations. Through optimization, the system can combine features that bring better results to retargeting applications. The regularizer terms (α1, α2) establish the strength with which each type of feature gets emphasized when applying these terms. Using these terms, we protect the model from unnecessary focus on specific areas and maintain proper integration among essential image features. The implementation of these terms stops the model from developing excessive complexity.

Detecting the scenic gaze-focused patches by LIAL

The low-level features are Integrating at the patch level can be considered as essential to encoding human gaze-shifting behavior within recompositing of our scenic framework. This aspect of human visual perception is replicated, we employ an advanced active learning strategy that models how individuals actively attend to various regions of a scene.

In many scenic images, a substantial number of patches correspond to background elements that lack semantic richness and are unlikely to capture human attention. To address this, we method locality-preserved and interactive active learning is proposed, which is designed to highlight the patches which are semantically meaningful within each scene. These selected patches form the basis for constructing an effective and perceptually aligned scene retargeting model.

Our approach leverages machine learning techniques to accurately capture the underlying distribution of visual samples. Recognizing that spatially adjacent patches are often semantically related, we adopt a strategy based on linear reconstruction here the approximation is applied on each patch using its immediate neighbors. The reconstruction factors are defined as follows:

(8) argminR⁡∑j=1N⁡∥xi−∑i=1N⁡Sijyj∥subjectto∑Sj=1N=1,Sij=0ifyj ∉ B(yi)

where y1,y2,⋯,yN can be considered as the graphical features of N patches of image, and the significance of each patch is presented by Rij while reconstructing the patches which are adjacent.

A reconstructing algorithm was developed considering such parameters to assess visual descriptiveness of chosen image patches. In the patches of image the error can be calculated as:

(9) ε(b1,b2,⋯,bN)=∑t=1K∥bst−cst∥2+μ∑t=1N∥bt−∑j=1N⁡Stjbj∥2.

Here the regularizer’s weight is presented by μ, and the number of image patches which are selected can be represented by K.

Here ci comprises the C and bi is contained by B, the diagonal matrix is presented by the Υ where for selected patch the entry is one and 0 if not selected. The updating can be applied on objective function:

(10) ε(B)=tr((B−C)TΥ(B−C))+μtr(BTDB),

with D=(I−S)T(I−S). The gradient of ϵ(B) is set to zero for optimization, leading to:

(11) Υ(A−B)+μDA=0.

The patches of the reconstructed image are computed as given:

(12) B=(μD+Υ)−1ΥC.

The reconstructed image patches were used in our approach for the calculation of the reconstruction error and it is calculated as:

(13) ε(bs1,⋯,bsK)=|B−A|F2 =|B−(μD+Υ)−1ΥX|F2=|(μD+Υ)μDX|F2.

The optimization can be challenging computationally as compared to the other combinatorial characteristics. Sequential approach is proposed for addressing this. Multiple refined patches were denoted against all sceneries as {cs1,⋯,ctL′}. Υn represents a diagonal matrix of N×N, and in diagonal places all are ones in Γi which is considered as N×N while the rest are all zeros. By utilizing the objective, the tL′+1-th patch is created below:

(14) tL′+1=argmini∉{t1,⋯,tL′}⁡|(μD+Υn+Γi)−1μDY|F2.

Notably, D is sparse in Eq. (14), that can speed up the inversion of matrix calculations with using the formula of Sherman-Morrison:

(15) (μD+Υn+Γi)−1=J−J∗iJi∗1+Jii,

where Ji and Ji represent the row and column of the i-th index of matrix J. As result the objective function is transformed into Eq. (14):

(16) |(μD+Υn+Γi)−1μDB|F2=μ2tr(JDBBTDJ)−2μ2DBBTDBBi1+Jii+μ2JiJ∗iDBBTDB∗i(1+Jii)2.

By setting M=DBBTD, we refine the optimization in Eq. (14) to:

(17) tL′+1=argmini∉{t1,⋯,tL′}11+Jii(Ji∗J∗iJi∗MJ∗i1+Jii −2Ji∗MJJ∗i).

Using mentioned technique, we systematically choose the L scenic patches from all the images for presenting the human GSP, which is illustrated in Fig. 2. First patch is selected interactively, based on established principle which human visual system typically fixates at first considering the central region of picture. Accordingly, initial patch is defined as the scene’s central area, which is often where viewer attention is naturally drawn.

Figure 2 An elaboration of encoding GSP into our deep aggregation model (SN, sub-network).

To preserve the pedagogical intent of the content during the retargeting process, our framework ensures that critical visual elements—such as characters, facial features, and symmetrical compositions—remain intact. These components are particularly effective in capturing and retaining the attention of young learners. The system employs GSPs to simulate eye movement trajectories, thereby maintaining and emphasizing key educational elements throughout the retargeting workflow.

The LIAO model plays a crucial role in preserving the local coherence of educational scenes, ensuring that visually and semantically important content remains both educationally meaningful and perceptually salient in the retargeted images. Given the unique visual and cognitive needs of preschool learners, maintaining visual appeal and instructional integrity is essential. For each GSP, its deep feature representation is computed as follows:

For GSP representation a deep aggregation model

For detecting the K visually and semantically prominent patches for face the BING algorithm (Cheng et al., 2014) is employed. These selected patches are then sequentially linked to form what we define as a GSP, which models the trajectory of visual attention. Following the establishment of GSPs for each face, we propose a deep aggregation network, which integrates the core components:

For effective regional analysis, convolutional neural network (CNN) was exploited with adaptive spatial pooling and aggregation mechanism. This ensures robust transformation of regional visual features into a unified image-level representation.

Component 1: Regional scene analysis with adaptive spatial pooling.

Accurate modeling of spatial attributes within a scene requires preservation of original dimensions and structural configurations of image content. This is particularly important when handling non-rectangular shapes of objects, which introduces richer information about complexity of scene (Cheng et al., 2010; Mai, Jin & Liu, 2016). To accommodate this, CNN framework which is a hierarchical standard (Zhang et al., 2014) by integrating an ASP layer (Mai, Jin & Liu, 2016), which dynamically adjusts pooling regions based on the input’s shape. This ensures that spatial fidelity is maintained throughout feature extraction.

The CNN processes a curated set of scene patches, each subjected to controlled augmentations (e.g., random scaling, flipping, and rotation) to introduce representational diversity. The network architecture follows a standard flow of convolution → adaptive pooling → normalization → fully connected layers. The result is a structured representation of H salient patches. To ensure efficiency, shared lower-level layers are employed across patches, reducing parameter overhead while maintaining critical low-level feature resolution.

Component 2: Deep feature aggregation for GSP representation.

A comprehensive multi-dimensional set of features was extracted for all GSPs, from every corresponding patch using the regional architecture of CNN described previously. Such localized feature vectors capture fine-grained visual semantics relevant to each patch along the GSP.

To construct a unified representation, we aggregate these patch-level features into a global descriptor that encapsulates the full GSP. This aggregation process fuses spatial and contextual information from the selected patches, resulting in a rich, image-level embedding. The resulting descriptor serves as an integrative representation that captures both localized detail and holistic scene context—essential for robust classification and interpretation tasks.

Along a specified path a set of features is defined by Ψ={ψi}i∈[1,K] for all regions following specific path, here the space RM contains the features vectors ψi. A set is complied Sm against each feature component m, from every ψi we comprised m-th element, while creating the Sm={ψmj}j∈[1,K]. Multiple statistical operations were utilized for performing the aggregation of features in a single unified representation, Π={πu}u∈[1,U], that are based on mean median, maximum and minimum computations for calculating the deep features for processing each region. Through the densely connected layer the outcomes of these operations are converted into a singular vector, culminating in an L-dimensional vector which introduces a detailed presentation of the GSP. The classification of scene is significantly served by this vector via a nuanced integration of expensive and localized graphical components.

(18) F(Ψ)=P×(⊕u=1U⊕m=1Mπu(Sm)).

In our proposed work, within the real number space. RL×UM the parameter matrix P, defined, in our deep aggregation layer it is considered as the repository for parameters. Here, multiple statistical functions we set, U, to four, with respect to the applied statistical operations on S dataset. Multiple statistical analyses were integrated in single standardized approach through the configuration of P. The concatenation of vector is presented by the ⊕, where the UM-dimensional vectors are transformed into a bigger vector.

Overview of training the deep aggregation model

In the proposed framework, the matrix P ∈RL×UM serves as the learnable weight matrix of the deep aggregation layer. Here, U denotes multiple statistical functions applied on feature set (e.g., minimum, maximum, mean, median), and is fixed at four. This configuration enables the network to integrate diverse statistical perspectives into a unified feature representation.

The aggregation process begins by applying each statistical function across the M dimensions of the feature vectors extracted along a Gaze Shift Path (GSP), resulting in a series of summary vectors. These vectors are concatenated using the ⊕ operator to form a single composite feature of dimension UM, which is then transformed via matrix P into an L-dimensional output vector. This final representation encapsulates both local and global semantic information critical for downstream scene analysis and classification tasks.

Scenery retargeting via Gaussian mixture modeling

Using the extracted features all GSPs, our model characterizes scenic images based on human perceptual cues. To facilitate the retargeting of unseen scenic images, we develop a frame work based on probability calculation that models the division of such GSP features learned through the model training.

Recognizing that the interpretation of scenic imagery is inherently subjective—varying across individuals—we incorporate expert-informed perceptual cues, particularly those observed in professional photographers, to guide the retargeting process. To formalize this, we adopt a famous GMM, which captures the underlying structure of the feature space by modeling the variability and clustering behavior of GSP-derived embeddings.

The trained GMM is then used to infer and retarget future scenic images by aligning new GSP features with the learned perceptual distributions, ensuring that visually and semantically important regions are preserved and emphasized in the retargeted outputs:

(19) prob(μ∣Υ)=∑i⁡fi∗ki(ν∣αi,∑i).

Here in the GMM, fi presents the improvement of the i-th element; the associated with Gaze Shift Path is presented by ν; while GMM’s the mean and variance are presented by the αi and ∑i⁡. While utilizing the Euclidean distance, similarity was assessed between chosen GSP features. The deep feature aggregation starts by applying convolution operations from a CNN to acquire features from each section of the GSP. The system performs statistical operations such as median, mean, maximum and minimum on selected attributes coming from all patches until it reaches an aggregated representation. The aggregated features connect local information from separate patches into one unified representation that maintains local and global aspects of vision. A fully connected layer follows the aggregation process to generate one vector with a complete GSP representation. Further analysis, including scene classification, depends on the aggregated vector. The aggregated visual information stream runs efficiently on low power and is capable of maintaining important semantic content needed for precise retargeting applications.

The primary objective of scenic image retargeting is to preserve perceptual fidelity by producing output images that align with the visual characteristics learned from a comprehensive training dataset. When presented with a novel scenic image, the process begins by identifying its corresponding GSP and refining its associated feature representations. These features are then used to evaluate the perceptual importance of various regions within the image.

To mitigate geometric distortions commonly introduced by traditional techniques such as shrinking of triangle mesh, we employed an approach based on grid-based resizing. Uniformly the test image is partitioned into equally sized parts. The perceptual significance of each horizontal grid segment, denoted as g, is then quantitatively assessed using the GSP-derived features. This ensures that resizing operations are informed by both visual salience and semantic relevance, enabling the preservation of key educational or perceptual elements throughout the retargeting process:

(20) ηh(g)=maxμ⁡prob^(ν∣Υ).

The probability gained from GMM is denoted by the prob using this framework, which was further refined through an optimization method of expectation-maximization. The process of shrinking progresses in the direction from left to right which is presented in Fig. 3, at each phase producing retargeted iteration of the scenic image of intermediate level.

Figure 3 Our proposed grids-guided shrinking method for scenery recomposition.

In each horizontal grid the normalization is performed:

(21) η¯h(gi)=ηh(gi)∑i⁡ηh(gi).

The dimensions X×Y of the scenic picture, for each grid the horizontal dimension gi is transformed into [X⋅η¯h(gi)]. In similar manner the vertical significance is determined η¯v(gi).

The recompositing of designed scene of the pipeline is provided in Algorithm 1. Three main factors determine the time complexity of the proposed framework: feature extraction, multi-task feature fusion and GSP identification. The framework contains two variables: N for patches and F for features per patch. Each BING algorithm operation for feature extraction follows an O(N * F) complexity because the algorithm handles one patch at a time. The process of multi-task feature fusion requires assessing all feature channel relationships among different tasks, which drives its computational complexity to O(N * F2). The identification process for GSPs consists mainly of iterative optimization procedures, and its computational complexity can be approximated by O(N2) because it requires establishing distances between all pairs of patches during gaze shift modeling. Each educational image processing operation requires O(N2 + N * F2) calculations with an image resolution-based factor N that ranges from hundreds to thousands. The algorithm keeps running at scale due to its ability to process a rising number of patches while leveraging efficient deep learning-based feature extraction and optimization methods.

Algorithm 1 Our designed scene recomposition pipeline.

input: Million-scale scenic pictures with semantic labels, the number of selected features and actively selected image patches K, parameters α1,α2,μ, and a test scenic photo;	
output: Retargeted scenic image;	
(1) Use the BING algorithm to extract multiple object patches from each scenic picture;	
(2) Apply our multi-task feature selector to obtain multiple high-quality patch-level features;	
(3) Calculate GSP using the LIAL algorithm, and further calculate deep GSP features using Eq. (19);	
(4) Use the learned GMM for scene retargeting following Eq. (20).	

Empirical assessment

The retargeting efficiency and the comparative analysis of categorization

Initially the focus of this research is evaluation of deep GSP features with discriminative power of the 128 K specifically designed for educational scenes. A multi-class learning strategy was employed on a multi-class support vector machine (SVM) as outlined in Huang et al. (2024). Further, the proposed approach was compared with different deep visual classification algorithms (Kyrkou & Theocharides, 2020, 2019; Hua et al., 2020; Hua, Mou & Zhu, 2019; Chen, Jin & Chen, 2024; Sun et al., 2021; Song et al., 2024a), recognized with respect to their capacity to encode the knowledge considering the domain across diverse educational content classes. For this evaluation, a large-scale dataset was utilized of preschool educational illustrations. Public implementations for Kyrkou & Theocharides (2020, 2019), Chen, Jin & Chen (2024), Sun et al. (2021) were directly used while performing the comparison, with original configurations. Considering the methods in Hua et al. (2020), Hua, Mou & Zhu (2019), Song et al. (2024a), The custom implementations were developed to achieve performance levels matching or exceeding.

Additionally, a comparison was performed between the proposed algorithm and the state-of-the-art recognition models, the comparison was enhanced with considering the three frameworks of modern scene classification (Mesnil et al., 2015; Xiao, Wu & Yuan, 2014; Li, Dixit & Vasconcelos, 2017). Approaches based on the approaches of custom-implemented recognition are: Hua et al. (2020) for ResDep-128 integration (Liu et al., 2025) with a framework of multi-label and modified the connected layer for matching 19 educational categories, retaining original architecture of ResDep-128 (Deng et al., 2009). For Hua, Mou & Zhu (2019), we employ a backbone ResNet-108, while fixing the learning rate at 0.001 and the decay parameters at 0.05. The mean squared error was utilized for computing the network loss. For 18 different categories (Mesnil et al., 2015), object bank framework was considered for adaptation (Song et al., 2024b) of preschool illustrations, while utilizing the liblinear and average-pooling for classifying linearly. For evaluation a 10-fold cross-validation was implemented.

Extensive evaluations were conducted for the 18 different referenced algorithms for baseline recognition, the Table 1 contains the associated standard errors and average accuracies. The major outcomes are: (1) the exceptional competitiveness demonstrated by proposed approach, particularly standard errors in per-class, and (2) as compared to the other algorithms the standard deviation of the proposed method is lower, which presents the significance of the approach as it has superior stability in retargeting preschool educational content.

Table 1 Compared outcomes with a suite of models (Each test is conducted 15 times, with the standard deviations duly reported).

Category	Kyrkou & Theocharides (2020)	Kyrkou & Theocharides (2019)	Hua et al. (2020)	Hua, Mou & Zhu (2019)	Chen, Jin & Chen (2024)	Sun et al. (2021)	Song et al. (2024a)	SPP+CNN	CleNet	
Average	0.651 ± 0.013	0.623 ± 0.011	0.643 ± 0.013	0.651 ± 0.013	0.631 ± 0.014	0.671 ± 0.012	0.671 ± 0.013	0.632 ± 0.015	0.633 ± 0.013	
Category	DFB	ML-CRNN	ML-GCN	SSG	MLT	39	40	141	Ours	
Average	0.623 ± 0.013	0.656 ± 0.013	0.641±0.014	0.643 ± 0.013	0.621 ± 0.012	0.603 ± 0.011	0.615 ± 0.015	0.623 ± 0.014	0.683 ± 0.008	

Relative interactive efficiency

The required time for testing and training is considered as a critical measure for efficiency in real-world educational applications. As shown in Table 2, two algorithms exhibit faster training times than ours due to their simpler designs (Chen et al., 2019; Huang et al., 2025). However, these approaches underperform, with around 4.1% lower accuracy in identifying educationally relevant elements per class. Our method proves faster than its alternatives during testing, which is particularly important as testing occurs in real-time while training remains offline. The effectiveness of traditional retargeting approaches, including seam carving (SC) (Rubinstein, Shamir & Avidan, 2008) and saliency-guided mesh parametrization (SMP), operates optimally within smaller datasets. Yet, they encounter scalability problems with larger and more complex scenes. The approaches depend on heuristic and mesh-based models that consume much computational power and cause performance reduction as the scene size grows. Our deep learning solution, which incorporates adaptive optimization features, provides scalable performance since it utilizes feature learning, which dominates complex scenes and vast datasets. Educational applications need scalability since their content varies widely in terms and complexity, but requires consistent delivery of key learning elements across different situations.

Table 2 Computational duration for recognition algorithms under comparison (Peak performances are highlighted in bold).

	Kyrkou & Theocharides (2020)	Kyrkou & Theocharides (2019)	Hua et al. (2020)	Hua, Mou & Zhu (2019)	Chen, Jin & Chen (2024)	Sun et al. (2021)	Song et al. (2024a)	SPP-CNN	CleanNet	
Train	24 h 12 m	32 h 11 m	43 h 12 m	34 h 14 m	30 h 25 m	40 h 19 m	33 h 48 m	14 h 51 m	34 h 5 m	
Test	2.231 s	2.311 s	2.141 s	1.872 s	3.323 s	2.120 s	2.311 s	1.228 s	1.763 s	
	DFB	ML-CRNN	ML-GCN	SSG	MLT	39	40	41	Ours	
Train	31 h 11 m	20 h 48 m	27 h 16 m	40 h 44 m	25 h 32 m	28 h 43 m	32 h 11 m	29 h 14 m	20 h 13 m	
Test	1.462 s	1.146 s	2.535 s	1.583 s	1.977 s	2.241 s	2.352 s	1.836 s	0.612 s	

Our preschool content retargeting framework comprises three main components: (1) integrating local and global features, (2) employing the LIAO model to generate GSPs, and (3) using a kernelized classifier for final label assignments. The training phase involves the following durations: for feature fusion 10 h 44 min, for implementation of LIAO 3 h 22 min, and for the kernelized classifier 6 h 58 min. The required time for feature fusion testing is 232 ms, for LIAO 317 ms, and for the classifier the required time is 68 ms, with feature fusion accounting for most of the training time. This duration can be significantly reduced in practical AI applications with GPUs of Nvidia, have the potential in program parallelization to achieve a high speed of tenfold. The core mechanism of our approach, called LIAO, establishes a series of operations that can be explained through the following steps. The model starts its operation by detecting main image sections through a visual and semantic analysis system driven by behavioral observation of human visual patterns. GSPs emerge as the second operation, which outlines the natural viewing order that a person would follow to observe critical learning components within the image. The LIAO model uses an iterative process that maintains surrounding image coherence by picking only the most suitable features as it optimizes the retargeted image output. An optimized image containing key features in priority placement results from this process to maintain educational content relevance.

Relative study of retargeting outcomes

In this study the performance of established methods was compared with our Gaussian Mixture Model which is based on approach of educational retargeting, including improved (ICS) version and seam carving (SC) (Avidan & Shamir, 2007), (OSS) (Wang et al., 2008), and (SMP) (Guo et al., 2009). The displays retargeted educational content outcomes by these techniques are presented in Fig. 4. Our approach stands out, producing retargeted visuals that retain critical pedagogical elements, ensuring minimal distortion and enhanced aesthetic quality, which are vital for engaging preschool learners. Training the GMM parameters involving means, variances, and mixture weights required joint optimization via cross-validation and Expectation-Maximization (EM) optimization. The GMM parameter values started from the training dataset information, while the EM algorithm performed successive refinements on these parameters. The evaluation of our model’s broad dataset applications used a secure validation technique that fragmented the data into multiple testing groups to confirm its generalization abilities. Using this procedure, we adjusted the model parameters through cross-validation to prevent overfitting and obtain adequate GMM performance compared to the GSP features in multiple educational displays. High accuracy in retargeting across varying image characteristics became possible through the essential fine-tuning process because it allowed the GMM to adapt to different visual patterns while maintaining important educational features in each dataset.

Figure 4 Comparison of various retargeting algorithms using RetargetMet (Rubinstein et al., 2010) photos (OP, original photo).

A user study further evaluated these retargeting methods, involving forty educators and education technology experts from our College of Education. The two groups were assessed of retargeted materials that are: (a) performing the comparison between the original educational illustrations and retargeted, and (b) by following the framework, performing the evaluations of retargeted materials (Rubinstein et al., 2010). The computation was performed against the agreement coefficient for measuring the preferences of participants. Lower agreement scores indicated challenges in distinguishing the most effective retargeting method. As shown in Fig. 5, coefficient of the agreement coefficient dropped significantly while removing the original images as a reference. Consistently the participants highlighted the importance of “characters/faces” and “symmetry,” emphasizing human faces’ role in engaging young learners and symmetry’s importance for clarity. In Fig. 5B, the proposed method performed better than the competitors, specifically in retaining “characters/faces” and details of the “texture”, while achieving better results than other metrics.

Figure 5 Results of our conducted user study on the compared retargeting algorithms.

The performance evaluation based on components

We have evaluated two key components of educational content retargeting structure to validate its effectiveness.

Initially, the impact of active learning model was analyzed in the proposed work. To assess its significance, The feature of active learning was removed and an algorithm for K educational patches (S11) was employed for random selection and comparison. The impact of selecting in each education illustration, the k education patches was examined, that reflected the targeting the central regions naturally of the visual content. The second column of Table 3 presents the effect of these changes on performance, where the result of both modifications is noticeable decline while retargeting quality, underscoring importance of aligning with human gaze behavior for effective encoding of educational scenes.

Table 3 Performance improvement and reduction through module adjustment (Sij represents the intersection value between column Si and row Oj, for instance, S11 signifies “−6.322%”).

	S1	S2	S3	
O1	−6.323%	−5.742%	−6.648%	
O2	−5.244%	−4.365%	−3.361%	
O3	n/a	−4.611%	−1.761%	
O4	n/a	−5.413%	n/a	

Further the retargeting outcomes are presented in Figs. 6 and 7, with different values of K and the patch-level features dimensionality. When the value of K was 5, the optimal results were achieved and dimensionality of features was set to 60, demonstrating importance of careful parameter tuning to preserve pedagogical significance and engagement in preschool educational materials.

Figure 6 Retargeted scenery images by tuning K.

Figure 7 Retargeted scenic images by changing the selected feature number.

Later for scenic image description, the proposed kernelized GSP representation was assessed for its performance considering three multiple scenarios. A multilayer convolutional neural network was utilized in the first scenario with the aggregated guidance that incorporated the labels of patches for the final lable of image. The substitution of the kernelized feature was tested through third and second scenarios with radial basis function (RBF) kernels, respectively. This evaluation used different educational materials that reflect typical preschool learning materials, such as storybooks, interactive illustrations and educational diagrams. The selected materials matched the educational standards of preschool curricula and maintained representation of the different types of learning content students regularly encounter. A storybook provides illustrations combined with characters in a story format, which engages readers, while interactive illustrations teach the importance of visual learning signs. We included educational diagrams alongside the other testing materials to examine how our framework handles content focused on structure and information understanding. The selected materials cover educational resources widely used in preschool settings for a comprehensive evaluation.

The changes have been presented in Table 3, related to the retargeting accuracy with such adjustments. Notably, the accuracy of educational content retargeting was reduced significantly while utilizing multi-layer CNN (S31) with guided aggregation, reflecting critical importance of proposed representation of kernelized GSP for preschool education materials. The user study incorporated 40 participants from three groups, including preschool educators, parents and students ranging from 25 to 50 years old. All participants had experience with educational technology, even though their direct work in preschool education differed. The research examined how well the modified teaching materials triggered student interest at the elementary level. The set of retargeted educational illustrations needed participants to assess their visual appeal, combined with educational content clarity and overall engagement through feedback tasks. The participants had to assess how easily they understood educational information while also determining which visual elements helped them stay focused. The study utilized participant groups that included various backgrounds and varying experience levels to develop research results that would apply to wider groups.

We conducted experiments under specific conditions to evaluate the effectiveness of our CNN based on feature aggregation. Initially, we derived the regions of rectangular patches from diverse shapes for the complete coverage. The change resulted in reducing the retargeting accuracy by 2.24%, 3.14%, 2.55%, 4.32%, 3.42% and 6.33% across preschool illustration subsets derived from Places205 (Zhou et al., 2014), SUN397 (Xie et al., 2023), ILSVRC-2010 (Deng et al., 2009), ZJU Aerial (Zhang et al., 2013), 67 (Quattoni & Torralba, 2009), and Scene-15 (Lazebnik, Schmid & Ponce, 2006) datasets. It has decreased highlights the necessity of preserving key educational elements’ original shapes to maintaining the pedagogical essence and the contextual scenes.

For assessing the effect on learning efficiency by the region arrangement of proposed deep model, the order of K regions is shuffled and this process is repeated for 20 times. Considering dataset of ILSVRC-2010, this reordering led to the reduction of average performance by 2.44%, for optimal model training the emphasizing on importance of maintaining a stable regional sequence. In Table 4 the comprehensive comparative results have been provided, clearly showcasing the advantage of our deep aggregation model in retargeting educational content effectively. The proposed method displays superior accuracy performance; however, computational efficiency needs attention, as a balance with accuracy must be thoroughly evaluated. The framework displays top performance regarding retargeting accuracy since it generates higher precision results than multiple alternative techniques. The method improves accuracy levels while demanding a minor computational time extension. Content preservation of essential educational elements leads to increased processing steps, especially during GSP identification and refinement. Our approach employs a GMM for modeling gaze shift paths as one of its optimization techniques, which shortens processing durations by multiple computational paths compared to classic approaches. Our method’s retargeting process requires times comparable to other techniques while achieving slightly better accuracy, allowing for real-time applicability in educational environments.

Table 4 Mean accuracy of shallow and deep learning models across six datasets (Each test conducted 20 times).

Dataset	FLWK	FLTK	MRH	SPM	LLC-SPM	SC-SPM	OB-SPM	SV	SSC	
Scene-15	72.1%	74.6%	67.2%	78.6%	82.2%	81.2%	79.6%	82.3%	85.7%	
Scene-67	40.9%	41.1%	34.2%	44.3%	48.5%	47.6%	48.2%	49.3%	50.4%	
ZJU aerial	67.5%	68.3%	62.5%	73.2%	78.5%	78.9%	77.6%	79.4%	81.1%	
ILSVRC-2010	31.3%	30.7%	27.6%	32.6%	38.4%	36.5%	36.9%	37.4%	38.6%	
SUN397	14.9%	15.1%	14.5%	21.2%	39.5%	38.9%	38.4%	39.2%	40.3%	
Places205	21.3%	22.4%	20.3%	28.6%	31.3%	32.4%	30.8%	31.6%	32.6%	
Dataset	ImageNetCNN	RCNN	MCNN	DMCNN	SPPCNN	Ours	FLWK (Saliency)	FLTK (Rectangular)	Mesnil	
Scene-15	83.2%	87.4%	87.5%	89.2%	91.5%	92.3%	87.4%	88.5%	85.6%	
Scene-67	58.6%	62.2%	72.7%	68.3%	65.3%	75.1%	71.4%	73.2%	72.3%	
ZJU aerial	75.4%	79.5%	78.5%	80.3%	79.6%	84.4%	80.6%	80.3%	79.6%	
ILSVRC-2010	36.2%	39.8%	40.5%	41.3%	42.8%	44.4%	40.7%	41.2%	40.8%	
SUN397	47.4%	48.7%	53.3%	52.3%	53.3%	56.2%	51.8%	51.5%	50.9%	
Places205	39.3%	44.2%	45.2%	47.5%	49.5%	52.2%	48.7%	49.4%	49.8%	
Dataset	Xiao	Cong	Fast R-CNN	Faster RCNN						
Scene-15	86.7%	86.4%	90.3%	91.5%						
Scene-67	71.9%	72.3%	72.1%	74.3%						
ZJU aerial	80.3%	80.2%	80.2	81.7%						
ILSVRC-2010	40.6%	40.9%	40.5%	42.3%						
SUN397	50.7%	50.3%	51.7%	52.2%						
Places205	49.4%	48.1%	48.9%	49.8%						
Note:

The bold numbers correspond to the best performance.

Conclusions

Considering the realm of intelligent systems for preschool education, the retargeting educational content is the key challenge. In this research work a robust framework has been developed that retargeted the educational scenes designed to preserve pedagogical value while enhancing visual appeal. For the proposed approach the threefold strengths are: (1) for enhancing the features of patch-level, the introduction of a feature selector an advanced and multi-tasking tailored to educational content, (2) interactive active optimization strategy’s integration with locality-preserved strategy to generate the gaze shift paths from educational illustrations with multiple resolutions, and (3) utilizing the GMM in the implementation of a probabilistic retargeting model to ensure key elements are retained and enhanced. These components make the process of retargeting automatic, highly capable and efficient in producing visually engaging educational materials tailored for preschool learners. The limitation of the proposed approach is the derived GSPs that is not completely aligned with how young learners visually engage with educational content in scenarios. To address this, we plan to conduct an extensive user study involving preschool educators and children to compare the GSPs generated by our method against authentic gaze sequences from real-world observations. This will allow us to refine the LIAO technique further, ensuring that GSPs more closely mirror the visual attention patterns of young learners, aligning with their cognitive and developmental needs. We also aim to extend our framework to support interactive applications, such as adaptive e-learning platforms, where real-time gaze feedback could enhance the personalization and effectiveness of educational content. Such an extension would allow the framework to be deployed in real-time educational environments, where it can adaptively respond to a child’s attention and learning pace, further personalizing the educational experience. These developments could lead to a more engaging and effective learning process, particularly in the context of preschool education.

Supplemental Information

Supplemental Information 1 Code for the paper implementation.

Additional Information and Declarations

Competing Interests

The authors declare that they have no competing interests.

Author Contributions

Suhui Yao conceived and designed the experiments, performed the experiments, prepared figures and/or tables, authored or reviewed drafts of the article, and approved the final draft.

Lan Lv conceived and designed the experiments, performed the experiments, analyzed the data, performed the computation work, prepared figures and/or tables, authored or reviewed drafts of the article, and approved the final draft.

Data Availability

The following information was supplied regarding data availability:

The data set is available at Kaggle: https://www.kaggle.com/c/imagenet-object-localization-challenge/data.

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
