# Peer review of "Integrated multi-task feature learning and interactive active optimization for scene retargeting in preschool educational applications"

_PeerJ Computer Science, doi:10.7717/peerj-cs.3035_

## Round 0.1 · original submission · Major Revisions

Dera authors
Your paper needs several improvements as suggested by the reviewers, so please carefully incorporate those ,also considering mine below , before resubmission


- Consider briefly defining **"adaptive image retargeting"** early on for readers unfamiliar with the term.

- The transition from preschool education to technical methodology feels abrupt.

- The **"Integrated Multi-task Feature Learning"** framework is introduced without sufficient motivation

- The **LIAO strategy** is novel but could benefit from a high-level intuition before diving into technical details

- Recomposing semantic segments of complex scenes is crucial for adaptive image retargeting" → Consider rewording more concisely

Reviewer 1 ·

Basic reporting

This study presents a hierarchical model for adaptive image retargeting in preschool education, emphasizing engaging and informative visual content. Using the BING objectness metric, the model quickly identifies important scene patches and integrates multi-channel features through an Integrated Multi-task Feature Learning framework. A novel strategy, Locality-Preserved and Interactive Active Optimization (LIAO), mimics human gaze behavior to construct Gaze Shift Paths (GSPs), enhancing scene understanding. Deep features from GSPs are aggregated and modeled with a Gaussian Mixture Model for effective retargeting. Experimental results show the proposed method surpasses five existing techniques, achieving 3% higher precision than the second-best method while using only half the testing time, demonstrating its efficiency and suitability for educational applications.
The paper presents promising results, but several aspects can be improved. Key details such as dataset characteristics, participant demographics, task design, and evaluation metrics need clearer documentation to ensure transparency and reproducibility. Additionally, explanations of algorithmic steps and parameter tuning should be more concise and complete. Enhancing grammatical accuracy and clarifying complex processes will further improve readability and overall impact.

Experimental design

The experimental design would benefit from a more systematic description of the experimental setup, including variable control, sampling strategy, and task consistency across trials. Introducing a baseline comparison group and reporting statistical significance measures would further strengthen the rigor and credibility of the findings.

Validity of the findings

The results look encouraging and support the main goals of the paper. However, to make the findings more trustworthy, it would help to include more details about how the experiments were done, how the results were measured, and whether the differences are statistically meaningful. Adding this kind of information would make it easier for others to believe and build on the work.

Additional comments

The paper can be further improved by incorporating the following comments.

1. The empirical evaluation section is solid, but more details on how the educational materials were selected and their real-world representativeness would enhance clarity.
2. The user study is valuable, but providing details on participant demographics and tasks would strengthen the findings and ensure result reproducibility.
3. The paper contains several grammar errors, such as "which demand a specialized solution," which should be "which demands a specialized solution." A thorough review of the paper for grammatical accuracy is recommended to remove such errors.
4. In the comparative analysis with other scene retargeting techniques, while the paper does a great job showing the advantages of the proposed approach, a more thorough discussion of the trade-offs between computational efficiency and accuracy would add depth to the evaluation.
5. The comparison with other algorithms in Section IV.2 could be expanded by discussing the limitations of these methods, especially in terms of scalability when dealing with more complex scenes or larger datasets.
6. The algorithmic complexity of the proposed framework could be better explained, especially in relation to the number of features and patches being processed. Providing a more detailed breakdown of time complexity for each step would be useful for practical deployment.
7. In the section where gaze shift paths (GSPs) are encoded into the deep aggregation model, a more concise explanation of the deep feature aggregation process would make it easier for readers to follow the steps involved in generating the final GSP representation.
8. The method used to evaluate the retargeting outcomes using GMM is effective, but it would be beneficial to discuss in more detail how the GMM’s parameters were fine-tuned during training. This would provide greater insight into how the model generalizes across different datasets.
9. The conclusion could highlight future directions, particularly the real-time application of this framework in interactive pre-school education and integration with adaptive learning technologies.

Reviewer 2 ·

Basic reporting

Feedback is given below

Experimental design

Feedback is given below

Validity of the findings

Feedback is given below

Additional comments

This paper makes a valuable technical contribution to intelligent educational systems through the development of a gaze-sensitive, patch-based image retargeting framework tailored for pre-school learning materials. It demonstrates solid technical depth, extensive experimentation, and appropriate use of benchmarks. However, there are key areas that limit the strength of the overall contribution:
• The manuscript uses mostly professional and academic English throughout. However, there are several instances where clarity could be improved.
• The paper provides a reasonable background in the areas of multi-task learning and scene retargeting. However, it would benefit from more comprehensive referencing, especially regarding prior applications of artificial intelligence in pre-school or early childhood education contexts.
• The manuscript follows a conventional structure suitable for PeerJ Computer Science. Figures are included and seem relevant, but some lack detailed captions or clarity in axis labels, making them harder to interpret independently.
• It lacks clear sectioning or transitions. It jumps from definitions to formulations and then into optimisation, without orienting the reader.

The research question is explicitly stated and revolves around enhancing educational image retargeting using gaze-based modeling and patch-wise optimization.
• The theoretical basis for how gaze alignment improves learning outcomes in children is implied but not empirically supported within the study.
• The educational impact is inferred rather than directly measured or validated.
• Lack of user-centric validation (e.g., feedback from children or teachers) weakens claims of real-world efficacy.
• The methodology section includes enough detail about architectural changes, feature dimensions, and experimental parameters for reproducibility.
• Some methods are described at a conceptual level without pseudocode or sufficient mathematical detail.
• Robust experimental design and data usage, but transparency in final output sharing could be improved.

---

## Round 0.2 · accepted · Accept

Dear authors
Thank you for your resubmission, based on the assessment from the expert's and my self, I'm pleased to notify that your manuscript is judged scientifically sound and is being recommended for publication. Congratulations and thank you for your fine contribution

Reviewer 1 ·

Basic reporting

The manuscript is improved as per suggestions and accepted for publication.

Experimental design

The changes made has improved the design of the manuscript.

Validity of the findings

The revised version has updated as per reviewer comments.

Additional comments

I appreciate the author in improving manuscript as per reviewer comments.